# Assessing the national essential medicines list selection processes: Instrument development and testing

Ekanki Saxena[1]*, Jillian C. Kohler[2], Kevin E. Thorpe[3], Nav Persaud[4]

1 Institute of Health Policy Management and Evaluation, University of Toronto, Toronto, Ontario, Canada, 2 Munk School of Global Affairs, University of Toronto, Toronto, Ontario, Canada, 3 Leslie Dan Faculty of Pharmacy, University of Toronto, Toronto, Ontario, Canada, 4 Nav Persaud, Department of Family and Community Medicine, St. Michael's Hospital, Toronto, Ontario, Canada

* ekanki.saxena@mail.utoronto.ca

## Abstract

### Background

National Medicines Policies use the National Essential Medicines Lists (NEML) to improve equitable access to medications. An effective list selection process should address a country's priority healthcare needs and be aligned with WHO guidelines.

### Objective

The purpose of this study was to develop and test an instrument to measure the effectiveness of NEML selection process design.

### Methods

An instrument consisting of 16 items, along with an associated rating scheme, was created through a literature review and consultation with subject matter experts. Reliability was assessed using intraclass correlation coefficients (ICCs) for absolute agreement and consistency. Validity was evaluated by comparing instrument scores with two proxy measures of NEML selection process effectiveness.

### Results

The total score for NEML selection process design effectiveness, based on ratings from 5 raters across 4 countries, demonstrated an absolute agreement ICC of 0.98 (95% confidence interval 0.91 to 0.99) and a consistency ICC of 0.97 (95% confidence interval 0.88 to 0.99). Instrument scores varied correspondingly with two proxy measures of NEML selection process effectiveness, indicating good validity.

**Data availability statement:** This study was designed as a pilot project. The data used to rate pilot study countries is available in the WHO repository, where NEMLs for each country are publicly available: https://www.who.int/teams/health-product-policy-and-standards/assistive-and-medical-technology/essential-medicines/national-emls The data underlying the instrument ratings can be found in the supplementary file. The corresponding author or the Unity Health Toronto Research Ethics Board (researchethics@unityhealth.to) can be contacted if additional data is needed.

**Funding:** The author(s) received no specific funding for this work.

**Competing interests:** The authors declare that they have no competing interests.

## Conclusion

The instrument developed in this study measures the construct of NEML selection process design effectiveness, which essentially evaluates the alignment of national policy content with policy intent. The instrument demonstrated good validity and excellent reliability.

## Introduction

Improving access to medicines is critical to attaining Universal Health Coverage (UHC), which is one of the World Health Organization's (WHO) Sustainable Development Goals (SDG 3.8). National Essential Medicines lists (NEMLs) are a key policy tool under the essential medicines concept. NEMLs are a shortlist of medicines selected to address the priority healthcare needs of a population and form the basis of medicine selection, supply, procurement, production, and donation [1]. The essential medicines (EM) concept is meant to promote health equity and improve global health by making safe, effective medicines accessible to all at affordable prices [2,3]. The WHO created its first model essential medicines list based on global health priorities in the 1970s, which has since been adopted and adapted by more than 150 countries [4–6]. Improving access to medicines is contingent on the NEML addressing the priority healthcare needs of a country, which in turn, is a direct result of the NEML selection process design that is in place. The NEML selection process design is, therefore, an important component of the policy [2,7].

The WHO serves as both an agency that generates norms and standards and an advocate for the EM concept globally [8]. The WHO publishes a model EML every two years, compiled based on global priority health needs, and provides a starting point for nations to begin their own EML selection process. Additionally, the WHO provides guidance documents on how to design an NEML Selection Process tailored to national needs, priorities, and contexts [1,6,7]. These guidelines serve as a global standard that embodies the intent of the EM concept and are central to preserving its values, principles, and processes.

The concept of effective policy design is critical in linking policy design inputs and outputs [9]. NEMLs are selected at the national level by a government body. The WHO provides procedural guidance on NEML selection and publishes an up-to-date model EML reflective of global health priorities. Effective policy design at the national level requires aligning WHO policy standards and guidance with the national EML selection process design to ensure that the intent (values, principles, and processes [6,7,10,11] of the EM policy is preserved. A properly designed selection process can ensure that NEMLs contain medicines to meet the priority healthcare needs of citizens and fulfill national policy objectives. The concept of effective policy design is critical to linking policy design inputs and outputs [9]. In this case, it requires aligning global policy standards and guidance with the NEML selection process design, in order to ensure that the intent (values, principles, and processes) [7,10,11]) of the EM policy is preserved.

There appears to be heterogeneity in NEML selection processes across WHO member countries, which often results in NEMLs not reflecting the intent of the EM policy [12]. Evidence from an observational study of 137 countries with an NEML found few associations between variations in the number and types of medicines on the NEMLs and country characteristics representative of healthcare needs [13]. This may indicate that NEML selection process design is based on factors other than priority healthcare needs [13].

The current literature focuses on understanding which features of the NEML selection process may contribute to the selection of an ill-suited NEML [13–20]. Some extant studies have compared NEMLs of WHO member countries and analyzed the differences against possible explanatory factors influencing the selection of medicines at the national level [13], examined how research and the availability of contextualized pharmacoeconomic data impact the selection process [15,16], and explored financial barriers that constrain the selection process [14,15]. These studies highlight the impact of national context on policy implementation and recommend changes to existing selection processes to align them with the WHO policy development guidance. As far as we know, no studies have specifically investigated the alignment of the NEML selection process design with global standards.

In this work, we seek to develop a new instrument that captures the concept of NEML selection process design effectiveness and allows for the quantitative evaluation of the alignment of the NEML selection process design with global standards.

## Methods

In this study we developed an instrument to measure NEML selection process design effectiveness using methods adapted from the literature and assessed the instrument's inter-rater reliability and validity using data from a pilot study sample [21,22].

### Instrument development

NEML selection process design effectiveness was operationalized as the extent to which existing NEML selection process content aligns with WHO NEML selection process development guidance documents, and the WHO EML selection process [3,7,22–28].

A list of items was generated to measure NEML selection process through a literature review by comparing the process to the values of an ideal selection process and against a framework of ethical priority setting. Documents relevant to the application of the Accountability for Reasonableness (AFR) framework and WHO NEML selection process development guidance were identified. We (ES) conducted a literature review up to August 2024. The following databases were searched: SCOPUS, Medline, Google, and Google Scholar. The terms searched included: *NEML, EML, EML selection criteria, NEML selection process, EML selection process, EML assessment, and EML selection process rating*. All documents published after the introduction of the EM concept (1975) were considered. All types of English-language literature were included (peer-reviewed articles, WHO repository documents, print articles, electronic articles, websites, books, and policy papers). Title and abstracts were screened for relevance. Purposive sampling was then applied to select literature and documents relevant to the development NEMLs and application of AFR for policy assessment.

Data was analyzed using modified thematic analysis. First, the WHO model EML selection process [3] and the WHO NEML selection process development guidance documents [3] were analyzed to identify ideal process values for an NEML selection process that is fully aligned with EM policy intent. These process values served as categories under which instrument items were generated, similar to methods described previously [29]. The process values were then deductively categorized: relevance, transparency, revision & appeals, and enforcement.

The response scale for each instrument item was established through a review of key policy documents from WHO member countries and relevant literature, including current and previous NEMLs and National Pharmaceutical Policies

(NPPs), associated legislation, relevant reports, and scholarly articles. All publicly available documents relevant to NPP development and NEML selection process design were included. Where available, the national health authority was contacted to provide official policy documents. WHO member countries with an NEML were sampled alphabetically until response saturation was reached. A modified Likert scale was used to generate a response scheme, with each item having a unique scale to capture the range of responses encountered in the policy document review. For each item, response options were arranged from least to most desirable and assigned a number rating to create a modified Likert rating scale. The highest rating for each item was based on the WHO model EML selection process design, as it is perfectly aligned with EM policy standards and should capture all features of an ideal selection process. The NEML selection process design effectiveness raw score was calculated by summing all item scores.

### Instrument testing

Instrument testing used quantitative methods to establish the reliability and validity of the newly developed instrument that measures NEML selection process design effectiveness. A pilot study assessing the NEML selection process design effectiveness of four WHO member countries was used to establish the instrument's reliability and validity. Four countries representative of different regions and with different types of documentation available were assessed in the pilot (Afghanistan, Papua New Guinea, South Africa, and Barbados) by stratifying countries examined during response scheme generation into income-level classifications and selecting one representative country from each category (See S1 File Table 1). As this instrument is examining a selection process it was deemed important to select countries representative of all resource settings. As well, this ensures that the instrument and response scheme can capture potential responses across economic groupings. Income level of a country is known to be a key determinant of medication access as it impacts affordability, availability and access [30]. NEML selection process design for each sample country was constructed

**Table 1. NEML selection process design effectiveness Instrument.**

| Item # | NEML Selection Process Design Effectiveness Evaluation |
|--------|--------------------------------------------------------|
| 1 | Explicit instructions for the selection of an expert committee exist. |
| 2 | The names, affiliations, and conflict of interest statements of expert committee members are publicly available. |
| 3 | The expert committee responsible for National Medicines List (NEML) selection operates with full scientific independence. |
| 4 | Detailed guidelines/principles for the expert committee to establish an essential medicines list exist. |
| 5 | Explicit and detailed selection criteria for essential medicines list selection exist. |
| 6 | There is explicit direction to base EM selection decisions on scientific evidence of efficacy and safety, as per the selection criteria. |
| 7 | The prevalence of health conditions and resistance patterns are considered in EML selection, as per the selection criteria. |
| 8 | The selection criteria of EMs explicitly considers financial implications when examining medicines with equal safety and efficacy. |
| 9 | The selection criteria of essential medicines assesses the feasibility of uptake (health care setting, personnel etc.). |
| 10 | There is clear evidence of a National Medicines Policy (NMP) explicitly emphasizing a focus on communication of NEML and clinical guidelines to the public and healthcare personnel. |
| 11 | The documentation associated with the decision-making process, such as meeting minutes, is made publicly available. |
| 12 | The selection process used to select EMs is published publicly. (Website, journal, industry paper, etc.) |
| 13 | There is a means for the public or other interested parties to question decisions on inclusion/exclusion of Essential Medicine on the NEML. |
| 14 | There are clear indications that the EM selection process is reviewed (external/internal review of information). |
| 15 | Selected EML Revised/Reviewed regularly (There are instructions to review/revise the NEML regularly). |
| 16 | The use and impact of EML implementation is monitored. (There are instructions to monitor the use and impact of the NEML as a policy tool.) |
| | **Total Rating** |
| | **Relative Rating** |

through a document review of NEMLs, NEML selection process documents, and NPP documents publicly available via government websites, organizations websites, and scholarly literature [31–33].

Four members of the research team (DM, LS, AB, IA) who had backgrounds in health services research and were familiar with NEMLs acted as the raters. All four raters were provided with relevant documents required to evaluate the sample countries and received training through an introductory presentation and job aid. Ratings were performed independently.

NEML selection process design scores were determined by applying the instrument to each sample country by five raters: four recruited raters and the author. The raters were given eight hours per country to independently perform a document review and assess NEML selection process design. Item scores were converted using a conversion factor so that all items were equally weighted out of five. The summation of converted item scores formed the total NEML selection process raw score (See S1 File Table 2 for conversion calculations). Total raw scores were converted to percentage NEML selection process design effectiveness. The mean and standard deviation of NEML selection process design effectiveness scores for each pilot study country were calculated. Statistical analyses were conducted using SPSS by the author.

### Reliability testing

Interrater reliability refers to the degree of variation in scores given by multiple raters for identical items [31]. Interrater reliability was determined using the Intraclass Correlation Coefficient (ICC) [31].

Two ICC forms were calculated [32]: two-way, random effects, consistency, multiple rater; and two-way, random effects, absolute agreement, single rater. ICCs and their associated 95% confidence intervals were calculated for total country scores and scores stratified by AFR conditions. Guidelines found in the literature were followed to design and report on interrater reliability for the instrument [31–33].

### Validity testing

Face and construct validity were established through iterative item generation, revision, and reduction in conjunction with subject matter experts from the WHO, academic specialists, and researchers.

In the absence of accepted measures of the NEML quality, we used two proxy measures: [1] the number of medicines recently added to WHO model list that were also recently added to the NEML (larger number indicates a responsive NEML process) and [2] the number of medicines recently removed from the WHO model list that remain on the NEML (smaller number indicates a responsive NEML process). Criterion validity was assessed by graphically comparing country NEML selection process design effectiveness mean scores against these two proxy measures for the pilot study sample countries.

### Ethics statement

This study used only publicly available policy documents and did not involve human participants, patients, or confidential data. As such, it did not require ethics review under our institution's policies.

## Results

### Instrument development

WHO guidance documents were reviewed, and recommended selection process design components, processes, and process values were identified. The identified components and processes formed 28 features of an ideal NEML selection process design. These 28 features were consolidated by logically combining related features into single questions, forming the basis of a preliminary instrument with 25 items (see S1 File Fig 1). In the first iteration, and in consultation with

**Table 2. AFR conditions mapped to process values and instrument items.**

| A4R Conditions | Definition | Process Values | Evaluation Criteria | Item# |
|---|---|---|---|---|
| Publicity | Decisions should be made on the basis of reasons (i.e., evidence, principles, values, and arguments) that fair-minded people can agree are relevant under the circumstances. Fair-minded people are defined simply as those who seek in principle to cooperate with others to find mutually justifiable solutions to priority-setting problems. | Transparency | The selection process used to select essential medicines is published publicly (website/journal/industry paper/etc.). | 12 |
| | | | The documentation associated with the decision-making process, such as meeting minutes, is made publicly available. | 11 |
| | | | The names, affiliation, and conflict of interest statements of expert committee members are publicly available. | 2 |
| | | | Explicit instructions for the selection of an expert committee exist. | 1 |
| | | Consultative | There is clear evidence of a National Medicines Policy (NMP) explicitly emphasizing a focus on communication of NEML and clinical guidelines to the public and healthcare personnel. | 10 |
| Relevance | Decisions and their rationale should be transparent and made publicly accessible. | Accountability | The expert committee responsible for NEML selection operates with full scientific independence. | 3 |
| | | | Detailed guidelines/principles for the expert committee to establish an EML exist. | 4 |
| | | Relevant Selection criteria & Evidence-based selection | Explicit and detailed selection criteria for essential medicines list selection exist. | 5 |
| | | | The selection criteria of essential medicines explicitly considers financial implications when examining medicines with equal safety and efficacy. | 8 |
| | | | There is explicit direction to base essential medicine selection decisions on scientific evidence or efficacy and safety, as per the selection criteria. | 6 |
| | | | The prevalence of health conditions and resistance patterns are considered in NEML selection, as per the selection criteria. | 7 |
| | | | The selection criteria of essential medicines assesses the feasibility of uptake (healthcare setting, personnel, etc.). | 9 |
| Revisions & Appeals | There should be opportunities to revisit and revise decisions in light of further evidence or arguments, and there should be a mechanism for challenge and dispute resolution. | Participation | | |
| | | Process Review & Revisions | Selected EML revised/reviewed regularly. (There are instructions to revise/review the NEML regularly.) | 15 |
| | | | There are clear indications that the essential medicines selection process is reviewed (external/internal review of information). | 14 |
| | | | There is a means for the public or other interested parties to question decisions on inclusion/exclusion of essential medicines on the NEML. | 13 |
| Enforcement | There should be either voluntary or public regulation of the process to ensure that the first three conditions are met. | Process Use & Implementation | The use and impact of EML implementation is monitored. (There are instructions to monitor the use and impact of the NEML as a policy tool.) | 16 |

subject matter experts, 12 items were removed based on the following: eight items were redundant, and one item focused on implementation rather than selection process design (see S1 File Fig 1). The 16 items of the new instrument (Table 1) were then mapped to WHO guidance process values (Table 2), which were further mapped to the AFR procedural fairness conditions (Table 2).

A rating scale was created for each item based on scenarios encountered during a document review of NEML selection process documents for 40 countries. Each of the 16 items had a unique rating scale, ranging between 2 and 6 potential scores. Each point on the rating scale represents a specific situation as defined by the rating scheme. Table 3 provides a detailed description of the rating scale and scheme.

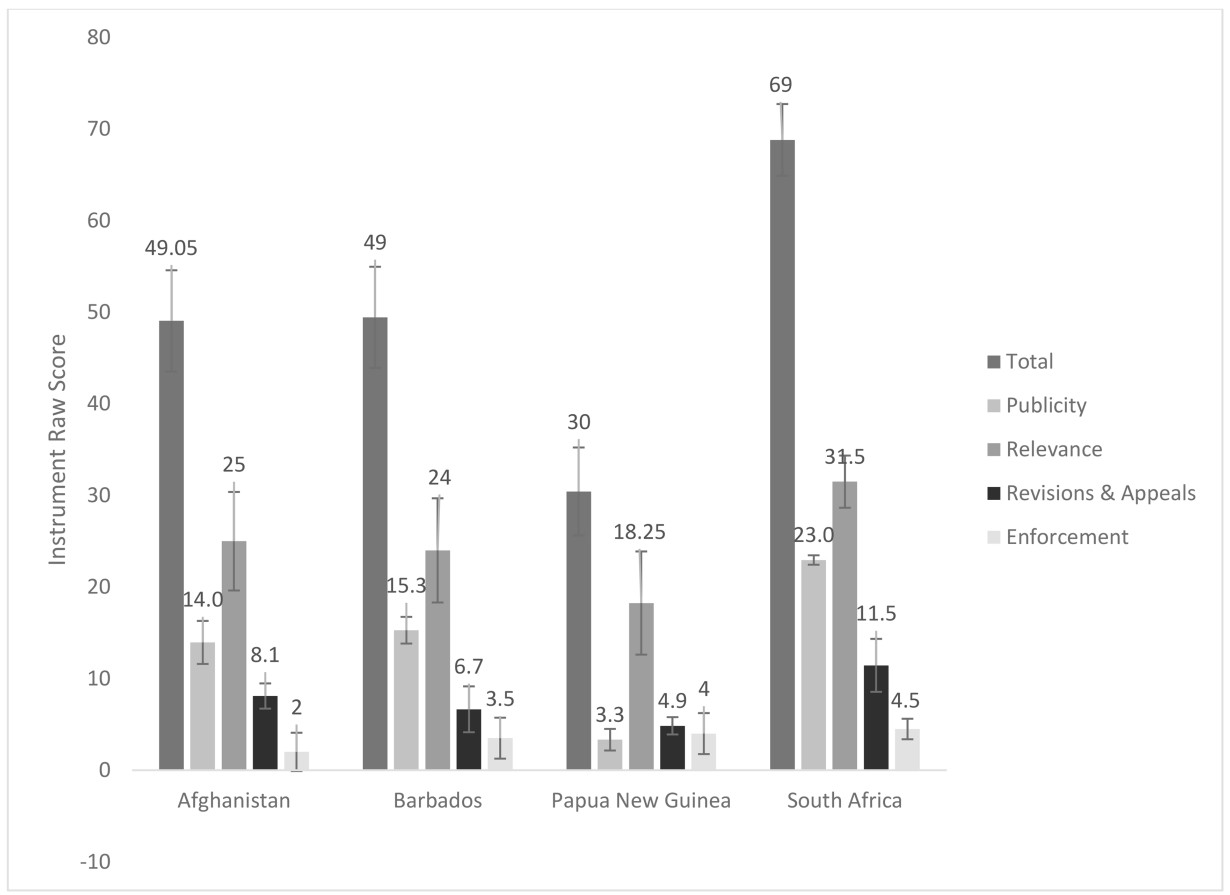

**Fig 1. Mean and standard deviation of total and AFR condition scores for NEML selection process design effectiveness in pilot study countries (Afghanistan, Barbados, Papua New Guinea, and South Africa).**

## Instrument testing

NEML selection process design effectiveness raw scores for each sample country were derived by summing the 16 converted instrument item scores, which are also presented as percentage scores (see Table 4). A perfect score on the instrument is 80. AFR condition scores for each country were calculated by stratifying instrument item scores as outlined in Table 1. The mean and standard deviation of scores for each country are also included in Table 4.

The highest-scoring country was South Africa, with a score of 68.8 (86%), followed by Barbados at 49.43 (62%), Afghanistan at 49.05 (61%), and Papua New Guinea at 30.43 (38%). Standard deviation indicates how much the data scatters around the mean scores or the degree of variation in rater assessments. Afghanistan and Barbados had the greatest standard deviations (5.5), followed by Papua New Guinea (4.8), while South Africa had the lowest variation (standard deviation 3.9).

The country scores for the Publicity condition followed a similar trend to the total score: South Africa scored highest (22.95), followed by Barbados (15.28), Afghanistan (13.95), and Papua New Guinea (3.33). Scores for the Relevance and Revisions & Appeals conditions differed slightly, as Afghanistan scored higher than Barbados. The Enforcement condition score was an exception, with Papua New Guinea [4] scoring higher than Barbados (3.5) and Afghanistan [2]. The similarities in score trends for Publicity, Relevance, and Revisions & Appeals conditions with the total score are evident in Figs 1 and 2.

**Table 3. Response scale and scheme.**

| | Evaluation Criteria | Rating Scale | 0 | 1 | 2 | 3 | 4 | 5 |
|---|---|---|---|---|---|---|---|---|
| 1 | Explicit instructions for the selection of an expert committee exist. | 0 - 5 | No process or explicit instructions for the selection of an expert committee exist. | A Process for selection of an expert committee exists, but is not documented. | An expert committee selection process with explicit instructions exists, and is documented; but it is only internally available. | An expert committee selection process with explicit instructions exists, is documented, and is published (publicly available). | An expert committee selection process with explicit instructions exists, is document, is published publicly. The process has rigorous oversight. | An expert committee selection process with explicit instructions exists, is document, is published publicly. The process has rigorous oversight, and there is accountability for selection. |
| 2 | The names, affiliations, and conflict of interest statements of expert committee members are publicly available. | 0 - 3 | No identification of expert committee members. | Names of the expert committee members are publicly available. | Names and affiliations of the expert committee members are publicly available. | Names, affiliations, and conflict of interest statements of the expert committee members are publicly available. | | |
| 3 | The expert committee responsible for National Essential Medicines List (NEML) selection operates with full scientific independence. | 0 - 2 | No initiative to ensure independence o2f expert committee. | Guidelines exist that acknowledge the scientific independence of the expert committee | Explicit implementation of scientific independence of the expert committee through legislation/policies/etc.. | | | |
| 4 | Detailed guidelines/principles for the expert committee to establish an essential medicines list exist. | 0 - 2 | No selection guidelines/principles exist | General guidelines/principles exist. | Detailed guidelines/principles exist. | | | |
| 5 | Explicit and detailed selection criteria for essential medicines list selection exist. | 0 - 4 | No selection criteria exists | An informal NEML Selection philosophy exists. | A formal but general NEML Selection philosophy exists. | A formal and detailed Selection Criteria for EM exists. | A formal and detailed Selection Criteria for EM,and is published for the public. | |
| 6 | There is explicit direction to base essential medicine selection decisions on scientific evidence of efficacy and safety, as per the selection criteria. | 0 - 1 | No explicit direction to link essential medicines selection to scientific evidence of efficacy and safety | Explicit direction to link essential medicines selection to scientific evidence of efficacy and safety | | | | |
| 7 | The prevalence of health conditions and resistance patterns are considered in NEML selection, as per the selection criteria. | 0 - 1 | No process or instruction to examine disease prevalence data during essential medicine selection. | Clear Process and explicit instructions to examine disease prevalence data during essential medicine selection. | | | | |

*(Continued)*

**Table 3.** (Continued)

| | Evaluation Criteria | Rating Scale | 0 | 1 | 2 | 3 | 4 | 5 |
|---|---|---|---|---|---|---|---|---|
| 8 | The selection criteria of essential medicines explicitly considers financial implications when examining medicines with equal safety and efficacy. | 0 - 1 | There are no guidelines or methods for assessing/handling the financial implications when examining medicines with equal safety and efficacy. | There is a method and guidelines for assessing/handling the financial implications when examining medicines with equal safety and efficacy. | | | | |
| 9 | The selection criteria of essential medicines assesses the feasibility of uptake (health care setting, personnel etc.). | 0 - 1 | No assessment of feasibility of uptake. | Assess feasibility of uptake – healthcare setting/personnel available etc. | | | | |
| 10 | There is clear evidence of a National Medicines Policy (NMP) explicitly emphasizing a focus on communication of NEML and clinical guidelines to the public and healthcare personnel. | 0 - 4 | No clear communication emphasized. | The NMP emphasizes a general philosophy to communicate NEML and clinical guidelines with few details/information available. | The NMP emphasizes an explicit instructions to communicate NEML and clinical guidelines with detailed information. | The NMP emphasizes explicit instructions to communicate NEML and clinical guidelines. There is clear evidence of multiple modes of communication, with detailed information. | The NMP emphasizes explicit instructions to communicate NEML and clinical guidelines. There is clear evidence of multiple modes of communication, with detailed information and avenues for general queries. | |
| 11 | The documentation associated with the decision-making process, such as meeting minutes, is made publicly available. | 0 - 2 | No documentation of decision making documents | EM decision making process documented, but not made publically available. | EM decision making process documented, and made publically available. | | | |
| 12 | The selection process used to select essential medicines is published publicly (website/journal/industry paper/etc.). | 0 - 3 | No Published Selection Process | Selection Process published internally only. | Selection process published publically. | Widespread publication and availability of Selection Process (website/journal/ etc.). | | |
| 13 | There is a means for the public or other interested parties to question decisions on inclusion/exclusion of essential medicines on the NEML. | 0 - 4 | No means or process exists. | Means or process exists to question decisions on essential medicines, however it is not clear/straightforward. | Clear and accessible process and means of questioning decisions on essential medicines inclusion/exclusion decisions. | Documentation of questions to decision makers and results/answers exists. | Documentation of changes made as a result of questioning of decision/ decision makers. | |

*(Continued)*

**Table 3.** (Continued)

| | Evaluation Criteria | Rating Scale | 0 | 1 | 2 | 3 | 4 | 5 |
|---|---|---|---|---|---|---|---|---|
| 14 | There are clear indications that the essential medicines selection process is reviewed (external/ internal review of information). | 0 - 5 | No review of selection process exists. | A general process exists for review of selection process. | A documented review process exists for internal review of selection process. | A documented review process exists for internal review of selection process, with regular process revision. | A documented review process exists for internal & external review of selection. | A documented review process exists for internal & external review of selection, with regular process revision. |
| 15 | Selected EML Revised/Reviewed regularly. (There are instructions to review/review the NEML regularly.) | 0 - 2 | No instructions to review/ revise NEML regularly. | Instructions to review/ revise NEML regularly, and evidence that NEML intermittently revised. | Instructions to review/revise NEML regularly, and evidence that NEML regularly revised. | | | |
| 16 | The use and impact of EML implementation is monitored. (There are instructions to monitor the use and impact of the NEML as a policy tool.) | 0 - 2 | No instructions to monitor use and impact of NEML. | There are instructions to monitor the use and impact of the NEML as a document and as a policy tool. (quantitative/ qualitative). | There are instructions to monitor the use and impact of the NEML as a policy tool (quantitative/qualitative). There exist published reports or articles evaluating the use and impact of NEMLs. | | | |

**Table 4. Converted out of 5 – Total, and AFR raw scores: mean and percentage, and standard deviation for pilot study sample countries.**

| Measure | Mean Score Statistics | Afghanistan | Barbados | Papua New Guinea | South Africa |
|---|---|---|---|---|---|
| Total | Score (%) | 49.05 (61%) | 49.43 (62%) | 30.43 (38%) | 68.8 (86%) |
| | Standard deviation | 5.5 | 5.5 | 4.8 | 3.9 |
| Publicity | Score (%) | 13.95 (56%) | 15.28 (61%) | 3.33 (13%) | 22.95 (92%) |
| | Standard deviation | 2.3 | 1.5 | 1.2 | 0.5 |
| Relevance | Score (%) | 25 (71%) | 24 (69%) | 18.25 (52%) | 31.5 (90%) |
| | Standard deviation | 5.4 | 5.7 | 5.6 | 2.9 |
| Revisions & Appeals | Score (%) | 8.1 (54%) | 6.65 (44%) | 4.85 (32%) | 11.45 (76%) |
| | Standard deviation | 1.4 | 2.5 | 0.9 | 2.9 |
| Enforcement | Score (%) | 2 (40%) | 3.5 (70%) | 4 (80%) | 4.5 (90%) |
| | Standard deviation | 2.1 | 2.2 | 2.2 | 1.1 |

## Reliability

Five raters independently assessed the four sample countries in the pilot study using the NEML selection process design effectiveness assessment instrument (see S1 File Table 3). Interrater reliability was assessed using two ICC forms selected based on established recommendations [30]: absolute agreement ICC using a two-way random effects model for single raters, and consistency ICC using a two-way random effects model for multiple raters [30]. Estimates of ICC and their 95% confidence intervals were calculated using SPSS statistical package versions 28 and 29 (SPSS Inc., Chicago, IL).

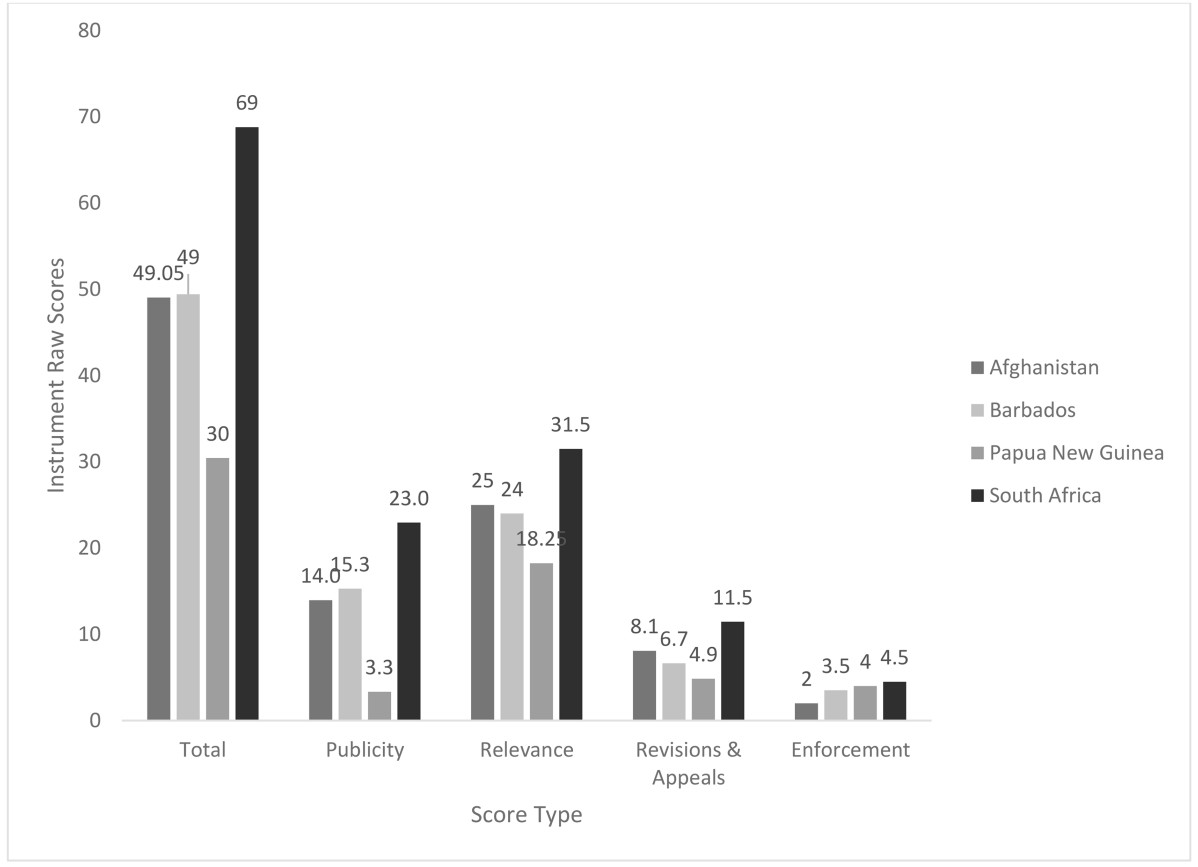

**Fig 2. Mean scores for NEML selection process design effectiveness across pilot study countries (Afghanistan, Barbados, Papua New Guinea, and South Africa), aggregated by score type (total and AFR condition scores).**

The NEML selection process design effectiveness total score showed an absolute agreement ICC of 0.98 (95% confidence interval: 0.91–0.99) and a consistency ICC of 0.97 (95% confidence interval: 0.88–0.99). The ICC for the Enforcement condition score was difficult to interpret due to there being only one item (See Table 5).

**Table 5. Intra-Class Correlation Coefficient estimates for NEML Assessment Instrument total score and AFR condition scores.**

| Measure | ICC Type | Intraclass Correlation | 95% Confidence Interval | |
|---|---|---|---|---|
| | | | Lower Bound | Upper Bound |
| **Total** | AA | 0.98 | 0.91 | 0.99 |
| | C | 0.97 | 0.88 | 0.99 |
| **Publicity** | AA | 0.99 | 0.97 | 1.000 |
| | C | 0.99 | 0.97 | 1.000 |
| **Relevance** | AA | 0.83 | 0.33 | 0.99 |
| | C | 0.84 | 0.28 | 0.99 |
| **Revisions & Appeals** | AA | 0.89 | 0.56 | 0.99 |
| | C | 0.89 | 0.53 | 0.99 |
| **Enforcement** | AA | 0.44 | −2.04 | 0.96 |
| | C | 0.43 | −1.90 | 0.96 |

Interrater reliability of NEML selection process design effectiveness scores aggregated by AFR conditions varied. The Publicity condition score showed excellent reliability: absolute agreement ICC (0.99, 95% CI: 0.97–1.00) and consistency ICC (0.99, 95% CI: 0.97–1.00). The Relevance condition score showed poor reliability: absolute agreement ICC (0.83, 95% CI: 0.32–0.99) and consistency ICC (0.84, 95% CI: 0.28–0.99), as the lower bounds of the 95% confidence interval were below 0.5. The Revisions & Appeals condition score showed moderate reliability: absolute agreement ICC (0.89, 95% CI: 0.56–0.99) and consistency ICC (0.89, 95% CI: 0.53–0.99), as the lower bounds of the 95% confidence interval were below 0.75. The confidence intervals for the Enforcement condition were wide, likely owing to the limited range in ordinal values. Due to varying AFR condition ICC values and wide confidence intervals, AFR sub scores should be used and interpreted with caution.

### Validity

Each component of the NEML selection process design effectiveness assessment instrument was developed in collaboration with specialists from the WHO. Face and construct validity were established by performing item generation, revision, and reduction iteratively in conjunction with subject matter experts (WHO, academic specialists, researchers). This collaboration process resulted in minimal revisions to the instrument items and rating scheme.

Instrument validity was also assessed graphically using scatter plots. Instrument scores were plotted on the x-axis against two external criteria on the y-axis: a proxy measure of an NEML selection process that is up-to-date (added medicines common to both global and national lists; Fig 3) and a proxy measure of an NEML selection process that is not up-to-date (medicines excluded from the WHO model EML but still on the NEML; see S1 File Table 5). Fig 3 shows that countries with higher instrument scores also had a greater number of medicines in common with the most recent additions

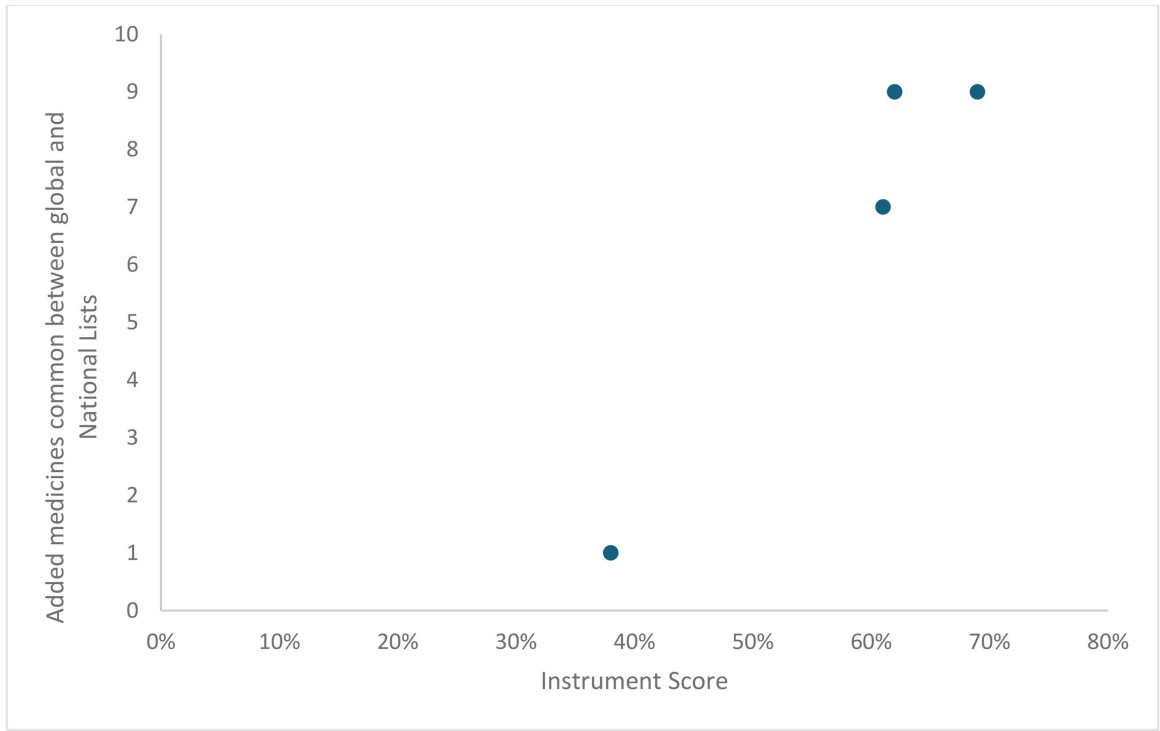

**Fig 3. Validity test results showing the relationship between instrument scores and a proxy measure indicating an up-to-date selection process across pilot study countries (Afghanistan, Barbados, Papua New Guinea, and South Africa).**

to the WHO model list. In contrast, Fig 4 shows that as country instrument scores increased, there were fewer medicines on their national lists that had been deleted from the corresponding WHO model list. Both findings align with our hypothesis and indicate that the instrument has good validity.

## Discussion

The NEML selection process plays a pivotal role in the overall policy process. A critical first step in designing and revising effective essential medicines policies to achieve health equity and improve population health is ensuring an effective NEML selection process design, one that is aligned with WHO standards, is in place. In this effort, we developed and validated a 16-item instrument for measuring NEML selection process design effectiveness. The instrument has good validity and excellent reliability overall.

Improving equitable access to medicines is the target outcome of Essential Medicines policies, as such there are existing methods that are aimed at monitoring progress. A well established and widely used indicator of access is the WHO/Health Access Internation medicines prices, availability, and affordability index [34]. Wirtz et.al. also propose five priority areas(Financing, Affordability, Quality and safety, Rational use, and Innovation) for policy evaluation in the journey towards Universal Health Coverage of essential medicines [30]. Bigdeli et.al. proposed a conceptual framework that embeds access to medicines in a health systems perspective accounting for supply and demand side barriers identified in previous frameworks. The proposed framework reorganizes components of the health system influencing access to medicines and accounting for the dynamic relationship between them [35]. To our knowledge however, this is the first attempt at a method to quantitatively evaluate a component of national medicines policy design that is based on recommendations

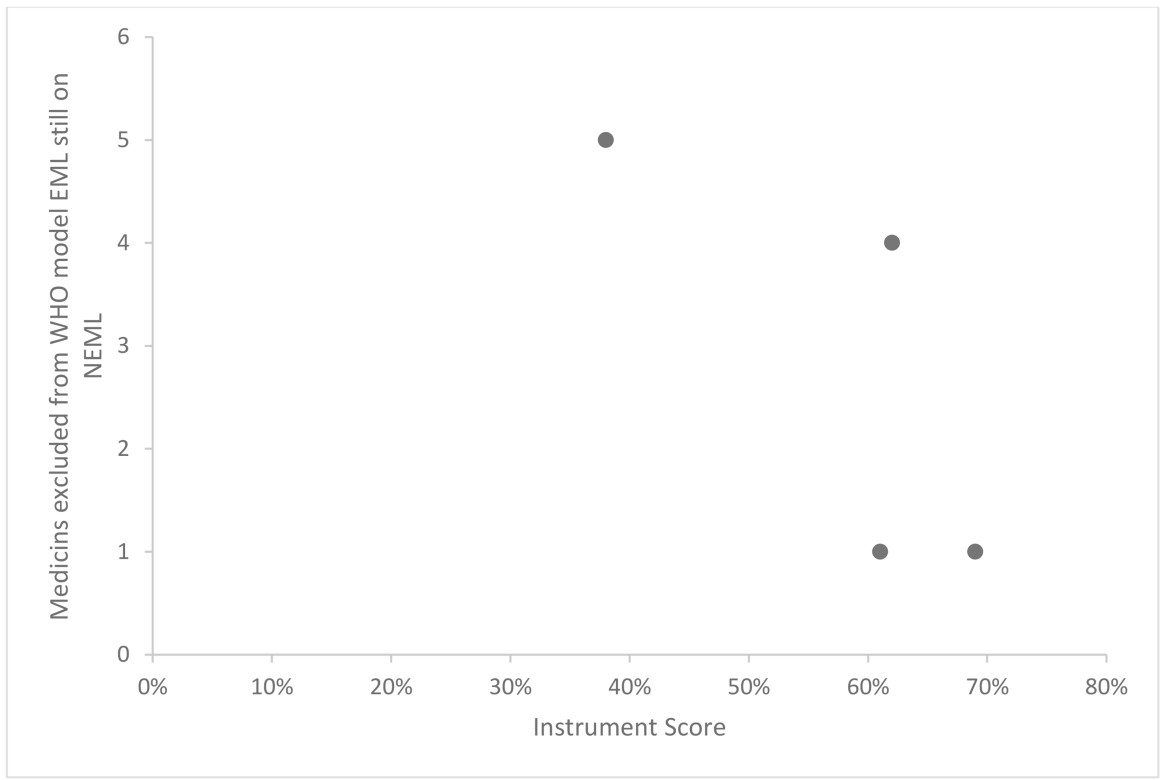

**Fig 4. Validity test results showing the relationship between instrument scores and a proxy measure indicating an out-of-date selection process across pilot study countries (Afghanistan, Barbados, Papua New Guinea, and South Africa).**

provided by the WHO. Currently, no validated instrument exists that quantitatively evaluates NEML selection process design effectiveness based on the assessment of policy content. Additionally, this is the first study to apply the AFR conditions to the evaluation of the NEML selection process design. Studies of the NEML selection process predominantly attempt to qualitatively examine the processes of individual countries [14–16,18,19], focusing on identifying process deficiencies. Quantitative inquiry has primarily focused on comparative analysis of NEMLs, which indirectly examines the NEML selection process design.[13,36] The IMF Institute for Health Informatics quantitatively compared medicines on the lists of nine countries and described the NEML selection processes of those countries.[20] The report classifies the country-level factors affecting list implementation into six categories: pricing, availability, reimbursement, government initiative, patent and licensing issues, and healthcare infrastructure. The impact of each category on the country is assessed on a negative-neutral-positive scale. Another quantitative observational study computed disparities among NEMLs, as well as between NEMLs and the WHO model EML for 137 countries.[13] The differences between lists were statistically analyzed against country characteristics reasonably expected to represent a population's healthcare needs [13], in order to identify factors that could explain the differences among national lists and discrepancies between NEMLs and the WHO model EML.

The WHO's efforts at aiding governance and advocacy for the EM concept globally have advanced the agenda of medication access [5,8], however, challenges still persist.[37] Prerequisite to ensuring the success of the EM policy in all four dimensions of access (availability, affordability, drug financing, and adequate supply) [7] is a selection process design capable of producing a national list that addresses the national health priorities. Several qualitative reports survey the components of national policy pertaining to the NEML selection process, alluding to its key role in the policy process. [38,39] This paper takes evaluation of EM policy further by ensuring the most effective design can be put in place, through the development of an instrument that quantitatively assesses the alignment of WHO policy standards with national policy design. The application of a valid and reliable instrument that evaluates NEML selection process design can be used not only to assess the current design of the NEML selection process, but also to provide direction on how to improve that design in low-scoring AFR categories.

## Limitations

Developing a new instrument is not without the challenge of establishing reliability of the measurement and validity of the instrument. Advantages of a new instrument include customizability to specific samples and data, both in items and range of responses. In this way, one can create a dynamic scale that accounts for the evaluation of real-world policies and responses. The NEML selection process design effectiveness assessment instrument developed here has excellent interrater reliability, along with reasonable interrater reliability of items, thereby providing meaningful information about the policy design of WHO member countries. However, it is recognized that country scores must be carefully interpreted, considering the low reliability of individual instrument items and Accountability for Reasonableness condition sub scores. This may mean that sub scores should be used only with caution and in context of the total score, and this is especially true for the enforcement domain which is based on a single item. As well, although the instrument validity was established on the basis of comparison with proxy measures of NEML selection process quality, it mainly assesses how quickly lists are updated to reflect changes made to the WHO model list. While timeliness of list update is an important dimension of process quality, it is acknowledged that it may not capture the concept of process quality in its entirety. While this is an important component of NEML selection process quality, it may not capture the full breadth of overall process quality. Additionally, despite efforts to mitigate variability in raters' interpretation of items and policy documents, this variability can ultimately impact ratings and interrater reliability. The instrument was piloted in four countries that are not necessarily representative. Furthermore, the reliance of the ratings on publicly available policy data carries the risk of bias, as countries without publicly available information may systematically differ (economically, geographically), and their situations may not be captured in the response scale.

## Conclusion

The instrument developed here measures the construct of NEML selection process design effectiveness, which examines the alignment of national policy content with policy intent. The final instrument has 16 items that were aggregated into the four AFR condition categories to help understand different contributions of the attributes of the NEML selection process design. The instrument has good validity and excellent reliability overall, however domains require further refinement. Future research could improve AFR condition sub score reliability by either increasing the number of items in each condition category or expanding the pilot study to include more countries and reassessing reliability.

This instrument provides a preliminary means to evaluate the policy content with respect to the NEML selection process design across countries. The data gathered can inform cross-country comparisons, assessments of scores over time, and to improve policy design based on both total scores and AFR condition sub-scores. Further validation in additional settings is needed to confirm broader applicability.

## Supporting information

**S1 File. Tables and Figures.**
(DOCX)

**S2 File. Ratings by each rater in pilot study.**
(XLSX)

## Acknowledgments

The World Health Organization (WHO) provided subject matter expertise and feedback during the instrument development process. The pilot country ratings were contributed by Darshanand Maraj (DM), Anjali Bali (AB), Itunu Adekoya (IA), and Liane Steiner (LS).

## Author contributions

**Conceptualization:** Ekanki Saxena, Nav Persaud.

**Data curation:** Ekanki Saxena.

**Formal analysis:** Ekanki Saxena.

**Investigation:** Ekanki Saxena, Jillian C. Kohler, Kevin E. Thorpe, Nav Persaud.

**Methodology:** Ekanki Saxena.

**Project administration:** Ekanki Saxena.

**Supervision:** Jillian C. Kohler, Kevin E. Thorpe, Nav Persaud.

**Writing – original draft:** Ekanki Saxena.

**Writing – review & editing:** Ekanki Saxena, Jillian C. Kohler, Kevin E. Thorpe, Nav Persaud.

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
