## [Decision Letter · Decision Letter 0]

4 Sep 2025

Dear Dr. Persaud,

Thank you for submitting your manuscript to PLOS ONE. After careful consideration, we feel that it has merit but does not fully meet PLOS ONE’s publication criteria as it currently stands. Therefore, we invite you to submit a revised version of the manuscript that addresses the points raised during the review process.

Deposit all data openly (mandatory for journal compliance).Provide ethics waiver/approval details.Revise the abstract, background, and structure as per R1’s points.Strengthen justification for country selection.Clarify and discuss weak reliability sub-domains.Tone down conclusions, add limitations/future work.Reformat appendices and improve figures/tables.Cite updated WHO sources.

We look forward to receiving your revised manuscript.

Kind regards,

Muhammad Shahzad Aslam, Ph.D.,M.Phil., Pharm-D

Academic Editor

PLOS ONE

Journal Requirements:

Additional Editor Comments:

Ethics: Need for clarification on waiver vs. approval (R1 #7–8). This is a compliance matter that must be resolved before publication.

Data availability: Both reviewers highlight that current practice (“on request”) does not meet PLOS ONE requirements. Must deposit data in an open repository such as OSF.

Sample size/generalizability: Both reviewers point out only four countries were included, which limits external validity. This cannot be fixed immediately, but must be framed transparently as a limitation.

Reliability: Some domains (e.g., Enforcement) show poor ICC. Both reviewers flag this. Authors must address it in discussion/limitations and ideally adjust the instrument.

Overstatement of conclusions: R1 is correct that claims are too strong given small sample + weak domains.

Appendices: R1 raises concerns that they are confusing, incomplete, or poorly explained. These can be corrected with careful revision.

Reviewers' comments:

Reviewer's Responses to Questions

**Comments to the Author**

1. Is the manuscript technically sound, and do the data support the conclusions?

Reviewer #1: Partly

Reviewer #2: Yes

2. Has the statistical analysis been performed appropriately and rigorously?

Reviewer #1: No

Reviewer #2: Yes

3. Have the authors made all data underlying the findings in their manuscript fully available?

Reviewer #1: No

Reviewer #2: No

4. Is the manuscript presented in an intelligible fashion and written in standard English?

Reviewer #1: Yes

Reviewer #2: Yes

Reviewer #1: I have reviewed the manuscript, but major comments need to be addressed accordingly, the manuscript addresses an important gap by proposing and validating an instrument to assess the design effectiveness of NEML selection processes. However, some methodological, conceptual, and reporting issues require attention before publication:

1. Title of study should be revised for better clarity, it should be catchy and concise

2. Background is long, make it short max 2 lines

3. Purpose/aim/objective of the study should be mentioned separately after the background.

4. The abstract reports ICC values but does not specify the pilot sample size (n=4 countries, 5 raters). Including this would improve transparency.

5. Key words should be mentioned alphabetically

6. Abbreviations should be written in full form first e.g. AFR, EML, NEML, NPPs etc, check this throughout the manuscript.

7. Ethics approval waiver letter is required, mention the details of waiver in revised manuscript and also provide the waiver letter to reviewers and editorial office for their perusal and consideration.

8. While the study used publicly available documents, the authors should clarify whether consultation with WHO experts and recruited raters required institutional ethics approval or exemption (particularly since human raters were involved). Currently, this section is minimal.

9. The authors claim this is the first validated instrument to assess NEML selection process design. While the novelty is clear, the manuscript could benefit from a deeper comparison with existing frameworks/tools for priority setting and policy evaluation e.g., WHO checklist approaches, health policy analysis frameworks. This would better situate the contribution.

10. Why Only four countries (Afghanistan, Papua New Guinea, South Africa, Barbados) were assessed? The rationale for their selection is only briefly explained (income-level stratification). A stronger justification is needed for why these specific countries were chosen and how representative they are. Small sample size also raises concerns about generalizability.

11. While total instrument reliability is excellent (ICC > 0.9), sub-domain reliability (e.g., Relevance, Revisions & Appeals, Enforcement) is weak due to wide CIs and limited items. The authors should discuss how this affects interpretability of subscale scores and whether additional items are needed to improve measurement stability.

12. It was observed that the proxy measures “alignment with WHO model EML additions/removals are reasonable but limited. They primarily capture timeliness rather than overall process quality. The authors should acknowledge this limitation more explicitly and suggest potential stronger gold-standard validity benchmarks.

13. The data availability statement indicates datasets are available upon request. This does not fully align with PLOS ONE’s data policy, which generally requires deposition in an open repository unless there are compelling reasons. Authors should ensure raw scoring data, item-level assessments, and country-level evaluations are made available.

14. The conclusion overstates the robustness of the instrument. Given some domains had low reliability and the pilot sample was limited, claims about wide applicability should be more cautious. The authors should emphasize that further validation in diverse contexts is necessary. Revise the conclusion accordingly.

15. Separately add the limitations and future recommendations section below the conclusion part, clearly mentioning the drawbacks of the current study, and what next can be done, discus this in a logical way.

16. The manuscript is generally clear, but some sections (Methods, particularly instrument development and response scale generation) are overly technical and could be streamlined for readability. A flowchart summarizing instrument development steps would help.

17. Table 3 (response scales) is long and dense; it may be better placed in supplementary materials. Figures 1–3 could use clearer captions that describe the key interpretation points (e.g., what higher/lower scores mean in context).

18. The manuscript uses “AFR” and “Accountability for Reasonableness” interchangeably. Standardize terminology to avoid confusion.

19. Several references are dated (e.g., 2001, 2002 WHO guidelines). More recent WHO documents (2023 model list and guidance) should be cited where applicable.

20. Appendix A, Pilot Study Sample Countries: Only four countries are included, which limits representativeness, the table does not explain why these specific countries (e.g., Afghanistan vs. other low-income countries) were chosen. Explicitly state the selection rationale beyond income-stratification (e.g., data availability, policy diversity). My recommendation is that discuss these points as a footnote or provide a brief explanation in discussion part.

21. Appendix B, Conversion Calculation: No worked example is provided, making it difficult for readers to follow how conversion works in practice. What was the logic behind choosing 42 as the total raw max score and converting to 80 is not sufficiently explained. I recommend to Include a step-by-step example for one item and country, showing raw score.

22. Appendix C, Explanations are superficial

23. Appendix D, Clarify the meaning of rater codes, add a legend, and consider averaging scores in a separate summary table.

24. Appendix E, Adds little beyond what is already in Results.Unclear whether these are averages across raters or consensus scores. State clearly if the totals are mean scores. If redundant with main text, move to supplementary-only.

25. Appendix F, Only percentages are provided; no statistical measures (correlation coefficients, regression outputs) are included. Proxy measures mainly capture timeliness of updates, not full process quality. Include statistical tests (e.g., Pearson/Spearman r, regression slopes). Acknowledge limitations of proxies more strongly.

26. Appendix G, Some domains show poor reliability (e.g., Enforcement ICC = 0.44, CI includes negative values). This is downplayed in the main text. The appendix does not explain why enforcement reliability is so weak (likely due to single-item measure). Add an explicit note on interpretability issues for domains with poor reliability. Consider revising the instrument to include more items under weak domains.

27. The appendices add value by making the methodology transparent, but they currently fall short of journal-quality supplementary material. They require: Better formatting (clearer tables, legends, consistent decimals), Stronger statistical reporting (validity beyond descriptive percentages). More detailed justification for methodological decisions (item reduction, country selection). Removal of duplicated or redundant sections. Without these revisions, the appendices risk confusing rather than supporting readers.

28. Major Revision Required: The manuscript is promising and fills an important methodological gap, but revisions are required to strengthen justification of the study design, address limitations of reliability/validity, clarify ethics and data availability, and moderate conclusions.

Reviewer #2: This study presents the first systematic development and validation of an instrument to evaluate the design effectiveness of National Essential Medicines List (NEML) selection processes, addressing a significant gap in the literature regarding quantitative assessment tools. The application of the Accountability for Reasonableness (AFR) framework to the evaluation of NEML selection processes is particularly innovative from a theoretical perspective. The study is well-designed, methodologically rigorous, and yields convincing results with strong theoretical and practical implications. It is recommended for acceptance after minor revisions, including expanding the sample size, optimizing the data availability statement, and elaborating on methodological details.

Specific Comments:

1.The instrument development process is rigorous, but the sample size is limited. The tool was developed based on WHO guidelines and expert consultation, demonstrating a systematic approach. However, the pilot study included only four countries, which may limit the generalizability of the findings. It is recommended to expand the number of countries in future studies to enhance the representativeness and robustness of the results.

2.Reliability is generally good, but certain sub-dimensions show low reliability. The ICC values for the total score indicate excellent reliability (>0.9). However, the ICC for the "Enforcement" condition is relatively low (0.44) with a wide confidence interval, suggesting poor inter-rater consistency for this dimension. Further refinement of the scoring criteria or additional items for this dimension are recommended.

3.The validity assessment approach is reasonable, but proxy measures require further justification. The use of "alignment with additions/removals on the WHO Model List" as a proxy measure is appropriate. However, the theoretical linkage between these indicators and "selection process effectiveness" should be more explicitly articulated to strengthen the validity argument.

4.The literature review and methods section could be more detailed. The description of the literature search strategy (databases, keywords, screening process) is somewhat brief. A more thorough methodological description is recommended to enhance reproducibility. Additionally, please clarify the rationale for retaining 16 items and indicate whether factor analysis or item response theory (IRT) was applied.

5.The current data availability statement does not comply with PLOS ONE’s requirements. The statement “Data are available from the corresponding author upon reasonable request” does not meet the journal’s requirement for full public data availability. It is recommended to deposit the data in a public repository (e.g., Figshare, Zenodo) and provide a DOI or access link.

6.Results are clearly presented, but figures could be improved. The figures (e.g., Figure 1–3) effectively communicate the main results but lack sufficient annotations and explanatory text. Please consider adding clearer labels, error bar interpretations, and indicators of statistical significance where applicable.

7.The policy implications are strong and practical. The developed instrument has clear value for cross-country comparisons, policy evaluation, and process improvement. It is suggested to further emphasize in the Discussion how the tool could be implemented by national or international organizations (e.g., WHO) and outline plans for future dissemination.

**Do you want your identity to be public for this peer review?** For information about this choice, including consent withdrawal, please see our Privacy Policy

Reviewer #1: No

Reviewer #2: **Yes:** Jing Zhang, Ph.D.

---

## [Author Response · Author response to Decision Letter 1]

10 Nov 2025

Reviewer comments

Reviewer #1: I have reviewed the manuscript, but major comments need to be addressed accordingly, the manuscript addresses an important gap by proposing and validating an instrument to assess the design effectiveness of NEML selection processes. However, some methodological, conceptual, and reporting issues require attention before publication:

1. Title of study should be revised for better clarity, it should be catchy and concise

***AUTHORS’ RESPONSE: We revised the title for clarity.

2. Background is long, make it short max 2 lines

***AUTHORS’ RESPONSE: We shortened the background section in the abstract as suggested.

3. Purpose/aim/objective of the study should be mentioned separately after the background.

***AUTHORS’ RESPONSE: We have added a separate objective section.

4. The abstract reports ICC values but does not specify the pilot sample size (n=4 countries, 5 raters). Including this would improve transparency.

***AUTHORS’ RESPONSE: We have added the sample size to the abstract.

5. Key words should be mentioned alphabetically.

***AUTHORS’ RESPONSE: The keywords have been put in alphabetical order.

6. Abbreviations should be written in full form first e.g. AFR, EML, NEML, NPPs etc., check this throughout the manuscript.

***AUTHORS’ RESPONSE: We have spelled out each term in its first instance.

7. Ethics approval waiver letter is required, mention the details of waiver in revised manuscript and also provide the waiver letter to reviewers and editorial office for their perusal and consideration.

***AUTHORS’ RESPONSE: This study used only publicly available policy documents and did not involve human participants, patients, or confidential data. As such, it did not require ethics review under our institution’s policies. We have revised the ethics statement to clarify this and we also share the waiver letter.

8. While the study used publicly available documents, the authors should clarify whether consultation with WHO experts and recruited raters required institutional ethics approval or exemption (particularly since human raters were involved). Currently, this section is minimal.

***AUTHORS’ RESPONSE: Consulting with experts was deemed by our institution not to require Research Ethics Board approval.

9. The authors claim this is the first validated instrument to assess NEML selection process design. While the novelty is clear, the manuscript could benefit from a deeper comparison with existing frameworks/tools for priority setting and policy evaluation e.g., WHO checklist approaches, health policy analysis frameworks. This would better situate the contribution.

***AUTHORS’ RESPONSE: We have revised the Discussion section to better situate this study in the literature.

10. Why Only four countries (Afghanistan, Papua New Guinea, South Africa, Barbados) were assessed? The rationale for their selection is only briefly explained (income-level stratification). A stronger justification is needed for why these specific countries were chosen and how representative they are. Small sample size also raises concerns about generalizability.

***AUTHORS’ RESPONSE: We have revised the methods and discussion section to indicate selected countries from different regions that we knew to have type of documentation available. We have discussed this as a potential limitation in the discussion section.

11. While total instrument reliability is excellent (ICC > 0.9), sub-domain reliability (e.g., Relevance, Revisions & Appeals, Enforcement) is weak due to wide CIs and limited items. The authors should discuss how this affects interpretability of subscale scores and whether additional items are needed to improve measurement stability.

***AUTHORS’ RESPONSE: We revised the manuscript to state in the limitations section of the discussion that the subscores should be used only with caution or in the context of the total score.

12. It was observed that the proxy measures “alignment with WHO model EML additions/removals are reasonable but limited. They primarily capture timeliness rather than overall process quality. The authors should acknowledge this limitation more explicitly and suggest potential stronger gold-standard validity benchmarks.

***AUTHORS’ RESPONSE: We have revised the limitation section of the discussion to indicate this.

13. The data availability statement indicates datasets are available upon request. This does not fully align with PLOS ONE’s data policy, which generally requires deposition in an open repository unless there are compelling reasons. Authors should ensure raw scoring data, item-level assessments, and country-level evaluations are made available.

***AUTHORS’ RESPONSE: We have revised this to indicate that the data used to rate the countries is available in the WHO’s repository. We can be contacted if additional data is needed.

14. The conclusion overstates the robustness of the instrument. Given some domains had low reliability and the pilot sample was limited, claims about wide applicability should be more cautious. The authors should emphasize that further validation in diverse contexts is necessary. Revise the conclusion accordingly.

***AUTHORS’ RESPONSE: We have revised the conclusion accordingly and moderated the claims.

15. Separately add the limitations and future recommendations section below the conclusion part, clearly mentioning the drawbacks of the current study, and what next can be done, discus this in a logical way.

***AUTHORS’ RESPONSE: We have added a separate limitations section. We have also discussed needed future work.

16. The manuscript is generally clear, but some sections (Methods, particularly instrument development and response scale generation) are overly technical and could be streamlined for readability. A flowchart summarizing instrument development steps would help.

***AUTHORS’ RESPONSE: We have revised the manuscript for clarity.

17. Table 3 (response scales) is long and dense; it may be better placed in supplementary materials. Figures 1–3 could use clearer captions that describe the key interpretation points (e.g., what higher/lower scores mean in context).

***AUTHORS’ RESPONSE: We have amended the captions for Figures 1-3.

18. The manuscript uses “AFR” and “Accountability for Reasonableness” interchangeably. Standardize terminology to avoid confusion.

***AUTHORS’ RESPONSE: We have standardized the terminology in the manuscript.

19. Several references are dated (e.g., 2001, 2002 WHO guidelines). More recent WHO documents (2023 model list and guidance) should be cited where applicable.

***AUTHORS’ RESPONSE: We have updated the manuscript with the most recent reference documents.

20. Appendix A, Pilot Study Sample Countries: Only four countries are included, which limits representativeness, the table does not explain why these specific countries (e.g., Afghanistan vs. other low-income countries) were chosen. Explicitly state the selection rationale beyond income-stratification (e.g., data availability, policy diversity). My recommendation is that discuss these points as a footnote or provide a brief explanation in discussion part.

***AUTHORS’ RESPONSE: We have addressed this comment in R1 comment 10 above.

21. Appendix B, Conversion Calculation: No worked example is provided, making it difficult for readers to follow how conversion works in practice. What was the logic behind choosing 42 as the total raw max score and converting to 80 is not sufficiently explained. I recommend to Include a step-by-step example for one item and country, showing raw score.

***AUTHORS’ RESPONSE: We have edited Appendix B to provide a worked example of the score conversion.

22. Appendix C, Explanations are superficial

***AUTHORS’ RESPONSE: We have revised the explanations in Appendix C and provided examples.

23. Appendix D, Clarify the meaning of rater codes, add a legend, and consider averaging scores in a separate summary table.

***AUTHORS’ RESPONSE: Appendix D amended

24. Appendix E, Adds little beyond what is already in Results. Unclear whether these are averages across raters or consensus scores. State clearly if the totals are mean scores. If redundant with main text, move to supplementary-only.

***AUTHORS’ RESPONSE: Amended table descriptions/caption to reflect their content

25. Appendix F, Only percentages are provided; no statistical measures (correlation coefficients, regression outputs) are included. Proxy measures mainly capture timeliness of updates, not full process quality. Include statistical tests (e.g., Pearson/Spearman r, regression slopes). Acknowledge limitations of proxies more strongly.

***AUTHORS’ RESPONSE: We have revised discussion/limitations section to acknowledge limitations of the proxy measures.

26. Appendix G, Some domains show poor reliability (e.g., Enforcement ICC = 0.44, CI includes negative values). This is downplayed in the main text. The appendix does not explain why enforcement reliability is so weak (likely due to single-item measure). Add an explicit note on interpretability issues for domains with poor reliability. Consider revising the instrument to include more items under weak domains.

***AUTHORS’ RESPONSE: We have revised the limitation section to indicate the issue with enforcement being a single item.

27. The appendices add value by making the methodology transparent, but they currently fall short of journal-quality supplementary material. They require: Better formatting (clearer tables, legends, consistent decimals), Stronger statistical reporting (validity beyond descriptive percentages). More detailed justification for methodological decisions (item reduction, country selection). Removal of duplicated or redundant sections. Without these revisions, the appendices risk confusing rather than supporting readers.

***AUTHORS’ RESPONSE: We have revised the appendices accordingly.

28. Major Revision Required: The manuscript is promising and fills an important methodological gap, but revisions are required to strengthen justification of the study design, address limitations of reliability/validity, clarify ethics and data availability, and moderate conclusions.

***AUTHORS’ RESPONSE: We have carefully revised the manuscript, and we believe the suggestions were very helpful.

Reviewer #2: This study presents the first systematic development and validation of an instrument to evaluate the design effectiveness of National Essential Medicines List (NEML) selection processes, addressing a significant gap in the literature regarding quantitative assessment tools. The application of the Accountability for Reasonableness (AFR) framework to the evaluation of NEML selection processes is particularly innovative from a theoretical perspective. The study is well-designed, methodologically rigorous, and yields convincing results with strong theoretical and practical implications. It is recommended for acceptance after minor revisions, including expanding the sample size, optimizing the data availability statement, and elaborating on methodological details.

Specific Comments:

1.The instrument development process is rigorous, but the sample size is limited. The tool was developed based on WHO guidelines and expert consultation, demonstrating a systematic approach. However, the pilot study included only four countries, which may limit the generalizability of the findings. It is recommended to expand the number of countries in future studies to enhance the representativeness and robustness of the results.

***AUTHORS’ RESPONSE: We have revised the manuscript to clearly state that scoring only four countries was a limitation. We have discussed the need for further work.

2.Reliability is generally good, but certain sub-dimensions show low reliability. The ICC values for the total score indicate excellent reliability (>0.9). However, the ICC for the "Enforcement" condition is relatively low (0.44) with a wide confidence interval, suggesting poor inter-rater consistency for this dimension. Further refinement of the scoring criteria or additional items for this dimension are recommended.

***AUTHORS’ RESPONSE: We have revised the limitations section to indicate the issue with subscores in general and the special issue with enforcement that is based on a single item. We have also revised the Conclusions section.

3.The validity assessment approach is reasonable, but proxy measures require further justification. The use of "alignment with additions/removals on the WHO Model List" as a proxy measure is appropriate. However, the theoretical linkage between these indicators and "selection process effectiveness" should be more explicitly articulated to strengthen the validity argument.

***AUTHORS’ RESPONSE: We have revised the Introduction and provided supporting references.

4.The literature review and methods section could be more detailed. The description of the literature search strategy (databases, keywords, screening process) is somewhat brief. A more thorough methodological description is recommended to enhance reproducibility. Additionally, please clarify the rationale for retaining 16 items and indicate whether factor analysis or item response theory (IRT) was applied.

***AUTHORS’ RESPONSE: We have provided additional details about the literature review.

5.The current data availability statement does not comply with PLOS ONE’s requirements. The statement “Data are available from the corresponding author upon reasonable request” does not meet the journal’s requirement for full public data availability. It is recommended to deposit the data in a public repository (e.g., Figshare, Zenodo) and provide a DOI or access link.

***AUTHORS’ RESPONSE: As we addressed in R1 comment 13 we have revised the data availability statement to indicate that the data used to rate the countries is available in the WHO’s repository. We can be contacted if additional data is needed.

6.Results are clearly presented, but figures could be improved. The figures (e.g., Figure 1–3) effectively communicate the main results but lack sufficient annotations and explanatory text. Please consider adding clearer labels, error bar interpretations, and indicators of statistical significance where applicable.

***AUTHORS’ RESPONSE: We have revised the figure captions to make them more clear.

7.The policy implications are strong and practical. The developed instrument has clear value for cross-country comparisons, policy evaluation, and process improvement. It is suggested to further emphasize in the Discussion how the tool could be implemented by national or international organizations (e.g., WHO) and outline plans for future dissemination.

***AUTHORS’ RESPONSE: Thank you. We have revised the discussion to discuss future work needed.

---

## [Decision Letter · Decision Letter 1]

28 Jan 2026

Assessing the National Essential Medicines List Selection Processes: Instrument Development and Testing

PONE-D-25-39957R1

Dear Dr. Persaud,

We’re pleased to inform you that your manuscript has been judged scientifically suitable for publication and will be formally accepted for publication once it meets all outstanding technical requirements.

Kind regards,

Muhammad Shahzad Aslam, Ph.D.,M.Phil., Pharm-D

Academic Editor

PLOS One

Additional Editor Comments (optional):

Reviewers' comments:

Reviewer's Responses to Questions

**Comments to the Author**

Reviewer #2: All comments have been addressed

2. Is the manuscript technically sound, and do the data support the conclusions?

Reviewer #2: Yes

3. Has the statistical analysis been performed appropriately and rigorously?

Reviewer #2: Yes

4. Have the authors made all data underlying the findings in their manuscript fully available?

Reviewer #2: Yes

5. Is the manuscript presented in an intelligible fashion and written in standard English?

Reviewer #2: Yes

Reviewer #2: The authors are to be commended for their thorough and thoughtful responses to the reviewers' comments. They have undertaken extensive revisions to the manuscript, which has significantly improved its clarity, rigor, and overall quality.

All major concerns raised in the previous round of review have been adequately addressed. Key improvements include:

A clearer title and abstract structure.

A more detailed justification for the pilot study country selection.

A revised data availability statement that now aligns with journal policy by directing readers to publicly available WHO repository data and supplementary files.

A standardized use of terminology and updated references.

A significantly strengthened discussion that better contextualizes the study within the existing literature.

A new, stand-alone "Limitations" section that candidly addresses the pilot sample size, the reliability of certain sub-dimensions (notably the single-item 'Enforcement' domain), and the inherent limitations of the proxy measures used for validity.

A more measured and cautious conclusion that accurately reflects the instrument's current stage of development as a promising but preliminary tool.

The revisions to the methods, results, and appendices have enhanced the transparency and reproducibility of the instrument development process. The authors have successfully moderated claims of robustness while highlighting the tool's potential utility for cross-country comparison and policy evaluation.

In summary, the manuscript now presents a well-developed and validated instrument that fills a clear methodological gap in the field. The authors have satisfactorily responded to all points raised, and the study meets the standards for publication in my view.

**Do you want your identity to be public for this peer review?** For information about this choice, including consent withdrawal, please see our Privacy Policy

Reviewer #2: No

---

## [Editor Report · Acceptance letter]

PONE-D-25-39957R1

PLOS One

Dear Dr. Persaud,

I'm pleased to inform you that your manuscript has been deemed suitable for publication in PLOS One. Congratulations! Your manuscript is now being handed over to our production team.

Kind regards,

on behalf of

Dr. Muhammad Shahzad Aslam

Academic Editor

PLOS One